# Projections of Future Climate Change in Singapore Based on a Multi-Site Multivariate Downscaling Approach

**Xin Li** [1],* , **Ke Zhang** [2],* and **Vladan Babovic** [3]

1   College of Hydrology and Water Resources and CMA-HHU Joint Laboratory for HydroMeteorological Studies, Hohai University, Nanjing 210098, China
2   State Key Laboratory of Hydrology-Water Resources and Hydraulic Engineering, CMA-HHU Joint Laboratory for HydroMeteorological Studies, and College of Hydrology and Water Resources, Hohai University, Nanjing 210098, China;
3   Department of Civil and Environmental Engineering, National University of Singapore, Singapore 119077, Singapore; vladan@nus.edu.sg
*   Correspondence: xinli@hhu.edu.cn (X.L.); kzhang@hhu.edu.cn (K.Z.); Tel.: +86-18851715807 (X.L.); +86-25-83787112 (K.Z.)

**Abstract:** Estimates of the projected changes in precipitation and temperature have great significance for adaption planning in the context of climate change. To obtain the climate change information at regional or local scale, downscaling approaches are required to downscale the coarse global climate model (GCM) outputs to finer resolutions in both spatial and temporal dimensions. The multi-site, multi-variate downscaling approach has received considerable attention recently due to its advantage in providing distributed, physically coherent downscaled meteorological fields for subsequent impact modeling. In this study, a newly developed multi-site multivariate statistical downscaling approach based on empirical copula was applied to downscale grid-based, monthly precipitation, maximum and minimum temperature outputs from nine global climate models to site-specific, daily data over four weather stations in Singapore. The advantage of this approach lies in its ability to reflect the at-site statistics, inter-site and inter-variable dependencies, and temporal structure in the downscaled data. The downscaling was conducted for two projection periods (i.e., the 2021–2050 and 2071–2100 periods) under two emission scenarios (i.e., representative concentration pathway (RCP)4.5 and RCP8.5 scenarios). Based on the downscaling results, projected changes in daily precipitation, maximum and minimum temperatures were examined. The results show that there is no consensus on the projected change in average precipitation over the two future periods. The major uncertainty for precipitation projection comes from the GCMs. For daily maximum and minimum temperatures, all downscaled GCMs project an increase of average temperature in the future. These change signals could be different from those of the original GCM data, both in magnitude and in direction. These findings could assist in adaption planning in Singapore in response to emerging climate risks.

**Keywords:** multi-site multivariate downscaling; Empirical Copula; inter-site correlation; inter-variable dependence; temporal structure; Singapore

## 1. Introduction

Estimates of the projected changes for the most relevant meteorological variables are essential for stakeholders to anticipate adaptation strategies for emerging climate risks. General circulation models (or global climate models) (GCMs) are often used to provide climate change information across the globe. However, as the spatial resolution of GCMs (with spatial resolution in the order of

several hundreds of kilometers) is too coarse to be directly used in regional or local climate change studies, downscaling approaches have to be developed to bridge the scale discrepancy between GCM grid-scale and the resolution required by regional or local studies. Dynamical downscaling (DD) and statistical downscaling (SD) are two types of approaches frequently used to downscale GCM outputs from GCM grid-scale to regional and local scales. The DD approach nests a regional climate model (RCM) (with spatial resolution in the order of tens of kilometers) into the parent GCMs, using GCM outputs as boundary conditions [1–3]. Although the DD approach is able to generate finer-resolution climate data, the scale discrepancy remains between the RCM grid-scale and the site-specific scale required by local studies. Besides, the RCM outputs are also subject to biases inherited from the parent GCMs as well as from their own model deficiencies [4]. In contrast to the DD approach, the SD approach is computationally easier because no physical representation or parameterization is involved, and only statistical links are established between the large-scale predictors and local-scale predictands. The SD approaches are generally categorized into three groups [4]: perfect prognosis (PP; also referred to as "perfect prog"), model output statistics (MOS) and stochastic weather generator (WG). For detailed reviews of these approaches, references are made to the works of Wilby and Wigley [5], Xu [6], Fowler et al. [7], Maraun et al. [4], and Schoof [8].

In the broad family of statistical downscaling, the WG approach has received widespread attention in recent decades due to its adaptability and flexibility in climate change studies. Weather generators are stochastic models which are able to generate ensembles of weather sequences with similar statistical properties as the observed one. Downscaling using WGs is done through perturbing the WG parameters according to the projected changes from climate models. This approach has been applied in downscaling both GCM and RCM outputs. For instance, Fatichi et al. [9] used an hourly weather generator to downscale the outputs from eight GCMs to the location of Tucson (AZ) for both the present and future period. Similarly, based on the change factor method, Kilsby et al. [10] developed a daily weather generator to generate site-specific meteorological series for the UK Climate Impacts Programme (UKCIP02) scenarios derived from the HadRM3H integrations.

Conventional downscaling approaches based on WGs are often single-site based, thus can only provide climate change scenarios at single locations or independently at multiple locations. The negligence of inter-site dependence in the downscaled meteorological fields could lead to misrepresentation of the extremes of hydrological response variables in subsequent hydrological impact studies. In addition, the inter-variable relationship between meteorological variables (e.g., the relationship between precipitation and temperature) should be respected to maintain the physical coherence in the downscaling procedure. As shown by Chen et al. [11], the ignorance of inter-variable correlation in the bias-correction of GCM outputs could lead to biases not only in the individual distributions, but also in their correlations, which in turn results in biased hydrological simulations.

All these issues necessitate the development of multi-site multivariate WGs (MMWGs)—with reliable representation of the inter-site and inter-variable dependence structures—for downscaling large-scale climate model outputs (please refer to Li and Babovic [12] for a detailed review of MMWGs). To fill such a research gap, efforts have been devoted to develop MMWGs with application to multi-site multivariate downscaling. Steinschneider and Brown [13] proposed a semi-parametric MMWG which integrates several components including a wavelet decomposition coupled to an autoregressive model to take account of low-frequency climate oscillations, a Markov chain and k-nearest-neighbor (KNN) resampling scheme to model the inter-site and inter-variable dependencies, and a quantile mapping procedure to allow for long-term distributional shifts resulting from prescribed climate changes. The proposed approach was shown to be effective in simulating spatially distributed, multivariate weather series for climate risk assessment. Chen et al. [14] proposed a hybrid approach which couples a spatial downscaling method and a temporal downscaling approach based on a MMWG (MulGETS, Chen et al. [15]). This approach was used to downscale the monthly precipitation from the National Centers for Environment Prediction (NCEP) data to daily precipitation data at ten stations over Hanjiang river basin in south-central China. As shown by Chen et al. [14], the proposed

approach was able to reconstruct the distributional statistics as well as the spatial dependence in the downscaled precipitation field for the validation period. However, given that the spatial correlation in MulGETS is taken into account by driving the single-site WG by temporally-independent but spatially-correlated random numbers based on the Iman shuffling method [16], the temporal persistence as well as inter-annual variability would not be maintained in the downscaled GCM outputs [12]. Likewise, Li et al. [17] presented a two-stage spatiotemporal downscaling approach, where, in the first step, spatial and temporal downscaling procedures were conducted based, respectively, on quantile mapping and a single-site Richardson-type WG, and then the inter-site and inter-variable dependencies in the downscaled GCM outputs were reconstructed using the Iman shuffling method as a post-processing technique in the second step. This approach is an extended application of the two-stage MMWG of Li [18]. However, as shown by Li and Babovic [12], the Iman shuffling method is incapable of reproducing the temporal structure in the post-processed series. To address this issue, Li and Babovic [12] proposed a two-stage MMWG based on Empirical Copula (EC) approach (MMWG-EC), in which the meteorological series are first simulated from a single-site multivariate WG, and then the inter-site and inter-variable dependencies, temporal persistence, and inter-annual variability are reconstructed using the EC approach. The proposed method was shown to outperform the one based on Iman shuffling, especially for restoring the temporal structures in the simulated meteorological series. To extend the proposed MMEG-EC approach to a multi-site multivariate climate downscaling context, Li and Babovic [19] proposed an integrated framework combining quantile mapping, stochastic WG, and the EC approaches for multi-site multivariate downscaling of global climate model outputs. The proposed method was applied to the Daqing River basin in north China, to downscale the monthly, grid-based GCM data to daily, station-based data at eleven stations over the catchment. As shown by Li and Babovic [19], the proposed approach performs reasonably well in restoring the marginally distributional statistics, temporal persistence, inter-site and inter-variable dependencies, and some extent of the inter-annual variability for the validation period.

Precipitation and temperature changes are complex in tropical urban regions such as Singapore, because the meteorological processes are influenced by the combined effects of natural climate variability, anthropogenic climate change and the local processes [20]. Precipitation in Singapore is often influenced by small-scale convective storms; thus, to model the change in precipitation extremes in future, downscaling approaches should be conducted at local scale, and at the same time the multi-site correlation should be preserved to reflect the actual spatial consistency and variability. Besides, as found by Li et al. [20], there are significant correlations between precipitation extremes and local temperature, which necessitates the use of a multi-variate downscaling approach in order to preserve the precipitation–temperature relationship in the downscaling procedure. Finally, to understand the spatial variability of the projected changes in precipitation and temperature in the Singapore region, a multi-site multivariate downscaling approach is desired. All these factors motivated this study.

Therefore, the primary objective of this study was to examine the projected changes in daily precipitation, maximum and minimum temperatures in Singapore, based on the newly developed multi-site multivariate downscaling approach of Li and Babovic [19]. To better characterize the uncertainty due to different schemes of GCMs, outputs from nine Earth System Models (ESMs) were downscaled. The projected changes in daily precipitation, maximum and minimum temperatures were examined, focusing on the changes in annual and seasonal means of precipitation and temperature. In addition, changes in average wet and dry spell lengths were also investigated.

## 2. Study Area and Observational Data

Singapore, a highly urbanized city-state with an area of around 719 km$^2$, is located off the southern tip of Malay Peninsula, separated from Peninsular Malaysia by Johor Straits to the north and from Indonesia's Riau Islands by the Singapore Strait to the south (see Figure 1). Singapore is situated one degree (137 km) north of the equator, and has a typically tropical climate with abundant

rainfall, high and uniform temperatures, and high humidity all year round. The long-term (1980–2010) mean annual precipitation is 2430 mm, with the average of 51% rainy days [21]. The highest daily precipitation depth in the historical record is 512.4 mm (recorded in December 1978) [22]. Annual mean temperature is around 26–27.5 °C [23]. Singapore's climate is characterized by two major monsoon seasons, i.e., the northeast monsoon season (from the end of October to early March) and the southwest monsoon (from June to September) [24]. Spatial variability of precipitation field is also observed for Singapore [25,26]. The long-term (1985–2014) daily precipitation, and maximum and minimum temperatures at four weather stations in Singapore (see Figure 1) were sourced from Meteorological Services Singapore (see http://www.weather.gov.sg/climate-historical-daily/). These three meteorological variables are often used as inputs to hydrological models; therefore, the investigation of their changes has great implication for further hydrological impact study.

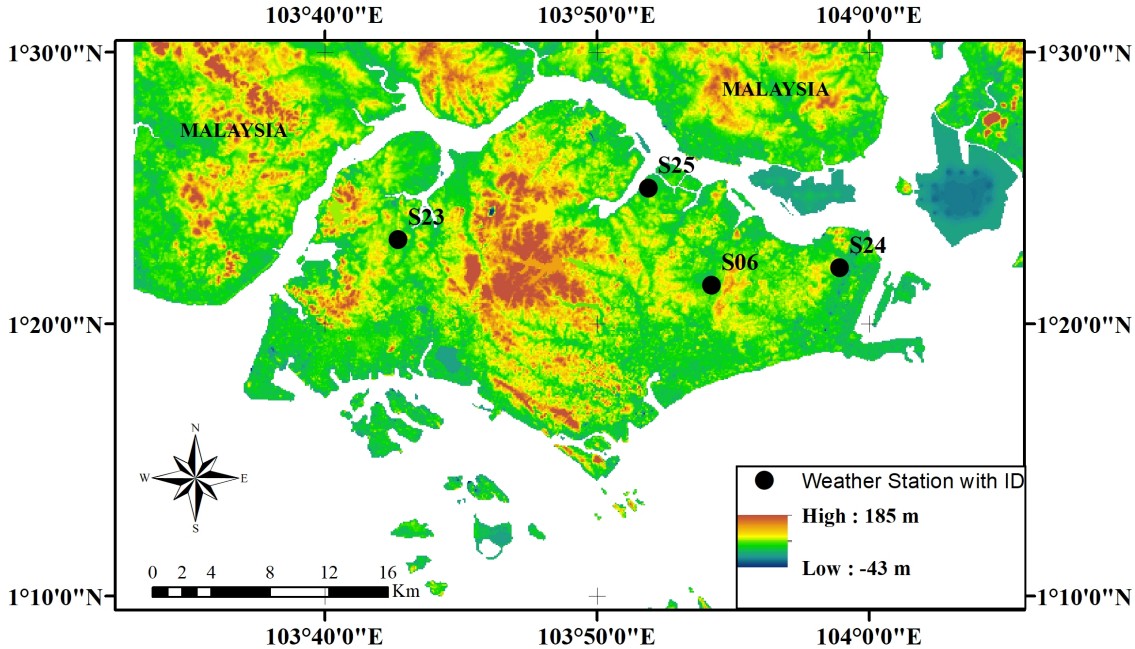

**Figure 1.** Location of weather stations in Singapore (elevation derived from the 90 m SRTM DEM [27]).

## 3. Methodology

### 3.1. Selection of Large-Scale Climate Models

Large-scale monthly model simulation and projection data from nine Earth System Models (ESMs, also referred to as GCMs) under historical forcing and two representative concentration pathways (RCPs) scenarios were used for downscaling. The two RCPs used are RCP4.5 and RCP8.5, which represent two emission scenarios under which the radiative forcing value in the year 2100 is approximately 4.5 and 8.5 W/m$^2$, respectively, higher than the preindustrial value. The nine ESMs were used by Marzin et al. [28] for Singapore's second national climate change study and are deemed to perform well in simulating the climate characteristics in Southeast Asia. It is necessary to emphasize here that, in the proposed downscaling approach, only the monthly GCM data are required. This renders the proposed methodologies more reliable given that monthly GCM outputs are more credible and readily available than those at daily scale [29]. The large-scale monthly GCM data were obtained from phase 5 of the Coupled Model Intercomparison Project (CMIP5, Taylor et al. [30]) model archive (http://cmip-pcmdi.llnl.gov/cmip5/). Since most of the GCM historical simulations end in 2005, to align with the historically observed record, historical simulation data for all the GCMs were extended to 2014 using their respective RCP4.5 simulations [31]. For the multi-site multivariate

downscaling, the historical period 1985–2014 was considered as the control period, during which the calibration of the methodologies was performed, and the periods of 2021–2050 and 2071–2100 were used as the mid- and end-century projection periods, during which the multi-site multivariate downscaling was conducted. The detailed information about the selected GCMs are given in Table 1.

**Table 1.** Large-scale climate models used for future climate change scenario construction.

| Earth System Models | Institution | Resolution (Latitude × Longitude) | Emission Scenarios |
|---|---|---|---|
| ACCESS1.3 | Commonwealth Scientific and Industrial Research Organisation, Australia (CSIRO), and Bureau of Meteorology, Australia (BOM) | 1.25° × 1.875° | historical, RCP4.5, RCP8.5 |
| Bcc-csm1-1-m | Beijing Climate Center | 2.7906° × 2.8125° | historical, RCP4.5, RCP8.5 |
| CanESM2 | Canadian Centre for Climate Modelling and Analysis | 2.7906° × 2.8125° | historical, RCP4.5, RCP8.5 |
| CMCC-CM | Centro Euro-Mediterraneo per I Cambiamenti Climatici | 0.7484° × 0.75° | historical, RCP4.5, RCP8.5 |
| CNRM-CM5 | Centre National de Recherches Meteorologiques/Centre Europeen de Recherche et Formation Avancees en Calcul Scientifique | 1.4008° × 1.40625° | historical, RCP4.5, RCP8.5 |
| CSIRO-MK3-6-0 | Commonwealth Scientific and Industrial Research Organisation in collaboration with the Queensland Climate Change Centre of Excellence | 1.8653° × 1.875° | historical, RCP4.5, RCP8.5 |
| GFDL-CM3 | Geophysical Fluid Dynamics Laboratory | 2° × 2.5° | historical, RCP4.5, RCP8.5 |
| HadGEM2-ES | Met Office Hadley Centre | 1.25° × 1.875° | historical, RCP4.5, RCP8.5 |
| IPSL-CM5A-MR | Institut Pierre-Simon Laplace | 1.2676° × 2.5° | historical, RCP4.5, RCP8.5 |

### 3.2. The Multi-Site Multivariate Downscaling Approach

The multi-site multivariate downscaling approach of Li and Babovic [19] consists of three steps. The first step is spatial downscaling, in which quantile mapping method is used to downscale monthly data from GCM grid-scale to site-specific scale. For the second step, temporal downscaling is conducted, in which the spatially-downscaled site-specific monthly data are further downscaled to daily data, by adjusting the parameters of a Richardson-type WG. In the last step, the EC approach is utilized to restore the observed inter-site and inter-variable dependencies as well as the temporal structures in the downscaled GCM outputs.

### 3.2.1. Spatial Downscaling

Before applying spatial downscaling, an inverse distance weighting method was used to interpolate the original large-scale GCM values at four nearest neighboring grid points to the grid box centered at the target station. The interpolation based on four nearest neighboring grid points is to avoid the risk of non-representative GCM data for point-scale climate change study [32,33].

A quantile mapping method, proposed by Zhang [34], was utilized to downscale the monthly GCM data from grid-scale to site-specific scale. This method has been applied in several studies (e.g., [14,17,32,35,36]), and is deemed to be effective in reproducing the cumulative distribution

functions (CDFs) of local observations in the downscaled data. First, the first- and third-order polynomials were fitted between the sorted (in ascending order) monthly data (monthly sum for precipitation and monthly means for maximum and minimum temperatures) of the observations and those of the GCMs for each calendar month (e.g., for January, the monthly data refer to data corresponding to January 1985, January 1986,..., January 2014) of the control period. Then, the fitted polynomials were used as transfer functions to downscale the GCM monthly data for the projection periods. More specifically, the third-order polynomials were used to downscale the monthly GCM values that are within the range in which the third-order polynomials were fitted, given their better goodness-of-fit than the first-order ones. For the out-of-range values, conservative approximations were generated based on the fitted first-order polynomials [14,34]. Mean and variance of the spatially downscaled GCM monthly data of the projection periods were calculated for each variable, each calendar month, and each station, which were used subsequently in the temporal downscaling scheme.

### 3.2.2. Temporal Downscaling

A Richardson-type WG [37] was used for temporal downscaling. Daily precipitation was simulated by a chain-dependent process, in which precipitation occurrence was modeled by a first-order two-stage Markov chain, and precipitation amount on a wet day (daily precipitation $\geq 0.1$ mm) was generated by a two-parameter Gamma distribution. For the modeling of daily maximum and minimum temperatures (Tmax and Tmin), the normal distribution of [17] was preferred, being it simpler than the first-order autoregressive model of Richardson [37]. Given the stochastic nature of the WG, an ensemble of 100 independent simulations was generated, each with a length equal to that of the projection periods. To downscale the spatially-downscaled GCM monthly data to daily data for the projection periods, parameters in the precipitation, maximum and minimum temperature models of the WG needed to be adjusted accordingly.

For downscaling of the precipitation, parameters needed to be determined for both the precipitation occurrence model (parameters being two transition probabilities) and precipitation amount model (parameters related to mean daily precipitation per wet day and daily precipitation variance for the wet days) for the projection periods. For precipitation occurrence, transition probabilities of the projection periods were determined based on a four-point linear regression method (regression between related precipitation occurrence model parameters and mean monthly precipitation for each calendar month and each station) calibrated using historical data [14,38,39]. For precipitation amount, the mean daily precipitation per wet day for the projection periods was calculated using the analytical approximation method of Wilks [40] based on the spatially-downscaled GCM mean monthly precipitation of the projection periods [19]. The daily precipitation variance was computed based on the proportional method of Zhang et al. [41], by multiplying the observed daily precipitation variance of the control period by the calculated variance ratios. The monthly variance ratios were calculated between the spatially-downscaled GCM monthly precipitation of the projection periods and the local monthly data of the control period for each month and each station.

Downscaling of GCM grid-scale monthly maximum and minimum temperatures for the projection periods involved perturbing the mean and variance of daily maximum and minimum temperatures for each calendar month and each station. The spatially-downscaled GCM mean monthly maximum and minimum temperatures of the projection periods were directly used as the adjusted mean daily values [34]. Similar to precipitation, adjusted daily temperature variances were determined by multiplying the observed daily temperature variances of the control period by the calculated monthly variance ratios (ratios between the spatially-downscaled GCM monthly temperatures and the observed local monthly temperatures of the control period).

### 3.2.3. The Empirical Copula Approach

The spatially and temporally downscaled GCM data (hereafter referred to as the "GCM-STD" data) were post-processed using the Empirical Copula (EC) approach to restore the observed temporal,

inter-site, and inter-variable dependencies. Empirical copulas [42–45]—with dependence structure defined by independent rank transformations of the samples in each of the dimensions of the multivariate data space–provide an ideal avenue for data post-processing. According to the definition of EC, if two multivariate data samples have identical rank structures, their empirical copula templates as well as all the information (including the temporal ordering characteristics and the inter-variable dependence structures) thereof encoded, are the same. That said, if we re-order a multivariate data sample according to the ranks in each dimension of some "reference" sample, the temporal ordering characteristics and multivariate relationships encoded in such "reference" sample are reconstructed. Moreover, being a post-processed technique that only manipulates the temporal ordering of the data points, all the marginally distributional attributes of the "to-be-processed" data points remain untouched.

A rank ordering technique adapted from the Schaake Shuffle method of Clark et al. [46] was used to post-process the "GCM-STD" data. The observed data of the control period were directly used as the "reference" to build the EC template and subsequently used to re-order the"GCM-STD" data, such that the observed temporal structures and the inter-site and inter-variable dependencies were reconstructed in the post-processed GCM-STD data (hereafter referred to as the "GCM-STD-EC" data) for the projection periods. To account for the effect of seasonality, the rank ordering method was performed separately for each of the combinations (3 variables × 4 stations in this study) for each calendar month (the number of data points for each month = number of years × number of days for that month; e.g., 30 × 31 = 930 for January in this study). For illustrative examples of applying the adapted Schaake Shuffle method, references are made to the works of Vrac and Friederichs [47] and Li and Babovic [12] in the contexts of multivariate, multi-site bias correction and multivariate, multi-site WGs, respectively.

The direct use of the observed data of the control period to build the EC template for the projection periods implicitly assumes the stationarity of EC—namely, the temporal, inter-site, and inter-variable dependencies remain constant, even in a non-stationary climate condition. In the climate downscaling community, the inter-variable and inter-site dependence structures (especially for the inter-site dependence) are often assumed to be stationary [17] or are treated as physical constraints [48] in a multi-site multivariate downscaling or bias-correction contexts. For the temporal dependence, the stationarity assumption might seem to be too strict, as temporal variabilities (from day-to-day variability to inter-annual variability) might change from the historical period to a future period.

To explore the stationarity assumption of EC based on observational data, Li and Babovic [19] compared three forms of dependence metrics including Lag-1 autocorrelation, inter-site Spearman correlation, and inter-variable Spearman correlation (which are manifestations of the ranks structures, or the EC template, of the data) between climate periods (the first-half 30-year calibration period and the latter-half 30-year validation period during 1957–2016) with different climatology for the Daqing River basin in north China. As shown, generally, these three forms of dependency metrics remain constant, with slight differences in Lag-1 autocorrelation and inter-variable correlation for some variables and variables pairs between the two periods. Moreover, the stationarity of these three dependence metrics in a non-stationary climate condition has also been found in the Upper Thames River basin in Canada [12]. Therefore, in the present study, we held the stationarity assumption of EC in the multi-site multivariate downscaling approach, given its merits in reproducing the multi-site, multivariate correlations, temporal persistence, and even some extent of the inter-annual variability in the downscaled GCM outputs [19].

## 4. Results and Discussion

### 4.1. Assessment of Inter-Site, Inter-Variable, and Temporal Dependence Reconstruction

The performance of the EC approach for reconstructing the observed inter-site Spearman rank correlation in the spatially-temporally downscaled meteorological field of ACCESS1.3 model under

RCP8.5 scenario over the period of 2071–2100 is evaluated in Figure 2. Similar performances were found for the remaining GCMs across the emission scenarios and projection periods; however, due to space limit, they are not presented here. For comparison, the results correspond to spatially-temporally downscaled outputs without applying the EC approach are also presented. The median values of inter-station correlations were taken over 100 different simulations and are compared against the observed ones of the control period. As shown, the spatially-temporally downscaled ACCESS1.3 data without adopting the EC post-processing technique (ACCESS1.3-STD) are not able to restore the observed inter-station correlations for all the meteorological variables. This is because, in the ACCESS1.3-STD data, the meteorological variables are downscaled based on a single-site spatial and temporal downscaling procedure, where no spatial links are introduced. In contrast, observed inter-station correlations are well reproduced in the spatially-temporally downscaled ACCESS1.3 data post-processed by the EC approach (ACCESS1.3-STD-EC) for precipitation occurrence, precipitation amount, and maximum and minimum temperatures at both daily and monthly timescales (daily precipitation occurrence and amount are summed up to monthly timescale, while daily maximum and minimum temperatures are averaged to monthly timescale). This is expected since the rank dependence structures of the observations are exactly honored in the ACCESS1.3-STD-EC data; thus, the inter-station correlations are preserved.

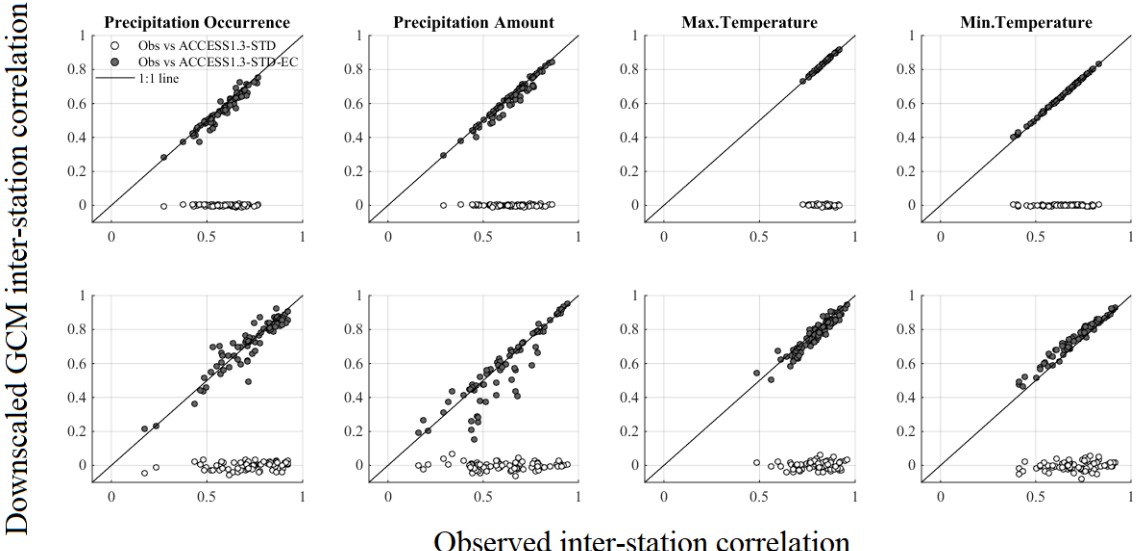

**Figure 2.** Inter-site Spearman rank correlations for precipitation occurrence, precipitation amount, maximum temperature, and minimum temperature for all possible station pairs and for all months for observations and spatially-temporally downscaled ACCESS1.3 data under RCP8.5 scenario over the period of 2071–2100. The four columns represent precipitation occurrence, precipitation amount, maximum temperature, and minimum temperature, respectively. The two rows denote the results for daily and monthly timescale, respectively. Median values across the 100 different simulations are shown against the observed values.

Figure 3 presents the Lag-1 serial autocorrelations (calculated as the Pearson correlation coefficients) for daily precipitation, maximum and minimum temperatures and the inter-variable cross correlations (computed as the Spearman rank correlation coefficients) for each pair of variables across the stations and months for both observations and the spatially-temporally downscaled ACCESS1.3 data under RCP8.5 scenario over the period of 2071–2100. Similar results are found for other GCMs across the emission scenarios and projection periods, and are not presented here due to space limit. As seen, the ACCESS1.3-STD data fail to represent the observed Lag-1 serial autocorrelations and inter-variable correlations. This is because the WG used in this study is not capable of representing the temporal persistence as well as the inter-variable relationships in the simulations. More specifically,

all the variables are modeled independently, and no autoregressive models are introduced to reflect the short-term persistence behavior. In comparison, the EC approach—by inheriting the rank structures from observation—successfully rebuilds the Lag-1 autocorrelation for daily precipitation, maximum and minimum temperatures, and the inter-variable correlations for all the variable pairs.

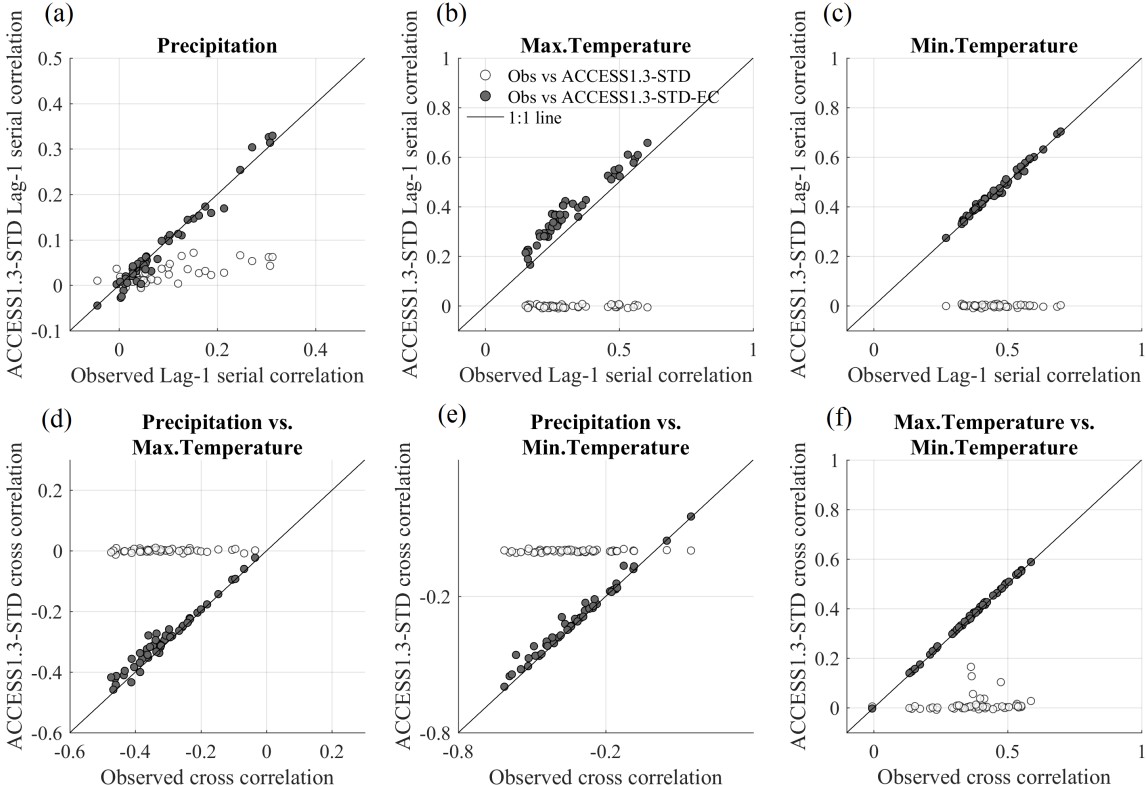

**Figure 3.** Lag-1 serial autocorrelations for daily: (**a**) precipitation; (**b**) maximum temperature; and (**c**) minimum temperature, as well as cross correlations (Spearman rank correlation) for: (**d**) precipitation vs. maximum temperature; (**e**) precipitation vs. minimum temperature; and (**f**) maximum temperature vs. minimum temperature, for all stations and months for observations and spatially-temporally downscaled ACCESS1.3 data under RCP8.5 scenario over the period of 2071–2100. Median values across the 100 different simulations are shown against the observed values.

### 4.2. Projected Climate Change

The proposed multi-site multivariate downscaling approach was applied to downscale precipitation, maximum and minimum temperatures from nine GCMs from grid-based, monthly scale to site-specific, daily scale at four weather stations in Singapore for the periods of 2021–2050 and 2071–2100 under RCP4.5 and RCP8.5 scenarios. The projected changes in annual and seasonal precipitation, annual average temperature, and wet and dry spell characteristics were examined. Box and whisker plots are utilized to summarize the results from 100 independent simulations, and the range of the box and whisker plot represents the stochastic variability of the downscaling approach. Noticing that, for each GCM, the downscaling result is ensemble-based, we focus on the sign of the median (positive or negative) and the range of the change values, which reflect, respectively, whether the change is more likely to be positive or negative, and the spread of the change. The statistical significance of the change for each realization of the downscaling in the ensemble is not considered, given that it would be complicated to draw a conclusion about the projected changes based on both the statistical significance and the spread of the changes.

4.2.1. Changes in Annual/Seasonal Precipitation and Temperature

Figure 4 presents the relative changes in average annual precipitation projected by downscaled precipitation outputs from nine GCMs under both RCP4.5 and RCP8.5 scenarios over the mid-century (2021–2050) and end-century (2071–2100) periods relative to observed values during 1985–2014 across the weather stations in Singapore. It should be emphasized here that the EC-component of the multi-site multivariate statistical downscaling method does not play any role for annual or seasonal means of precipitation and temperature, since it only operates on the temporal ordering of the downscaled data. As can be seen, there is no general agreement with projections of change among the GCMs. Both positive and negative changes in annual precipitation could happen in the future, and the magnitude of the changes depend on the selected GCM, emission scenario, projection period, and station of interest. Although the projected changes differ between different periods and scenarios, the major uncertainty comes from the GCMs. Note that, for most of the GCMs, the projected median change values are positive (more than half of the 100 realizations project positive changes) for both RCP scenarios and projection periods, suggesting that the increase of annual precipitation is more likely. This is consistent with the observed changes found by Beck et al. [25], Li et al. [26], and Li et al. [20], showing that annual precipitation as well as annual wet-day precipitation totals have increased significantly over the historical period. In addition, although the distances between these stations are relatively small, the projected changes are different across the stations. For instance, for the 2071–2100 period under RCP8.5 scenario, changes of mean annual precipitations are projected to vary between −15.9% and 30.8%, between −13.4% and 29.0%, between −16.6% and 33.7%, and between −16.9% and 31.7%, respectively, for stations S06, S23, S24 and S25 across the GCMs. These projected ranges widen further if the changes are assembled across the stations and GCMs. Changes of mean annual precipitations in Singapore are projected to range between −5.4% and 20.6% and between −9.5% and 25.9% over the mid-century period under RCP4.5 and RCP8.5 scenarios, respectively, across the stations. For the end-century period, mean annual precipitation could change by from −4.3% to 28.9% and from −16.9% to 33.7%, respectively, under RCP4.5 and RCP8.5 scenarios. Note that the range of the projected ranges are assembled rather than averaged across the weather stations, hence they represent the whole ranges of the projected changes over different weather stations in Singapore.

Figures 5 and 6 present, respectively, the relative changes in average northeast monsoon and southwest monsoon precipitations. Similar to annual precipitation, the projected changes in these two monsoon seasonal precipitation vary among the GCMs. Overall, average northeast monsoon precipitation is projected to change by from −16.0% to 27.9% and from −19.5% to 29.2% over the 2021–2500 period under RCP4.5 and RCP8.5 scenarios, respectively, across the stations. For the 2071–2100 period, the projected changes in average northeast monsoon precipitation range between −14.3% and 30.0% and between −22.1% and 32.5% under the RCP4.5 and RCP8.5 scenarios, respectively. In comparison, the spreads of the projected changes in average southwest monsoon precipitation under RCP4.5 and RCP8.5 scenarios, respectively, are between −15.4% and 41.8% and between −29.7% and 50.9% over the 2021–2500 period, and between −13.9% and 54.8% and between −31.5% and 77.8% over the 2071–2100 period.

For temperature, Figures 7 and 8 show, respectively, the changes in mean annual average daily maximum and minimum temperatures projected by the GCMs under both RCP4.5 and RCP8.5 scenarios for the periods of 2021–2050 and 2071–2100 relative to observed values of 1985–2014. Unlike precipitation, all the GCMs agree with the increases of average daily maximum and minimum temperatures for both projection periods and for both emission scenarios. The changes in average daily maximum and minimum temperatures could be felt differently at different stations. For instance, for the end-century period under the RCP8.5 scenario, average daily minimum temperature is projected to increase by 1.4–3.2 °C and 3.1–6.5 °C across the GCMs, respectively, for stations S23 and S06. Across the stations, average daily maximum temperature are projected to increase by 0.6–1.6 °C and 0.6–1.7 °C for the period of 2021–2050 under RCP4.5 and RCP8.5 scenarios, respectively. For average daily minimum temperature over the period of 2021–2050, the projected increases are 0.4–1.8 °C and

0.4–2.0 °C, respectively, under RCP4.5 and RCP8.5 scenarios. The increases become even greater into the future, with a projected increase of 0.9–3.0 °C (under RCP4.5) and 2.0–5.7 °C (under RCP8.5) for average daily maximum temperature, and 0.6–3.6 °C (under RCP4.5) and 1.4–6.5 °C (under RCP8.5) for average daily minimum temperature over the period of 2071–2100.

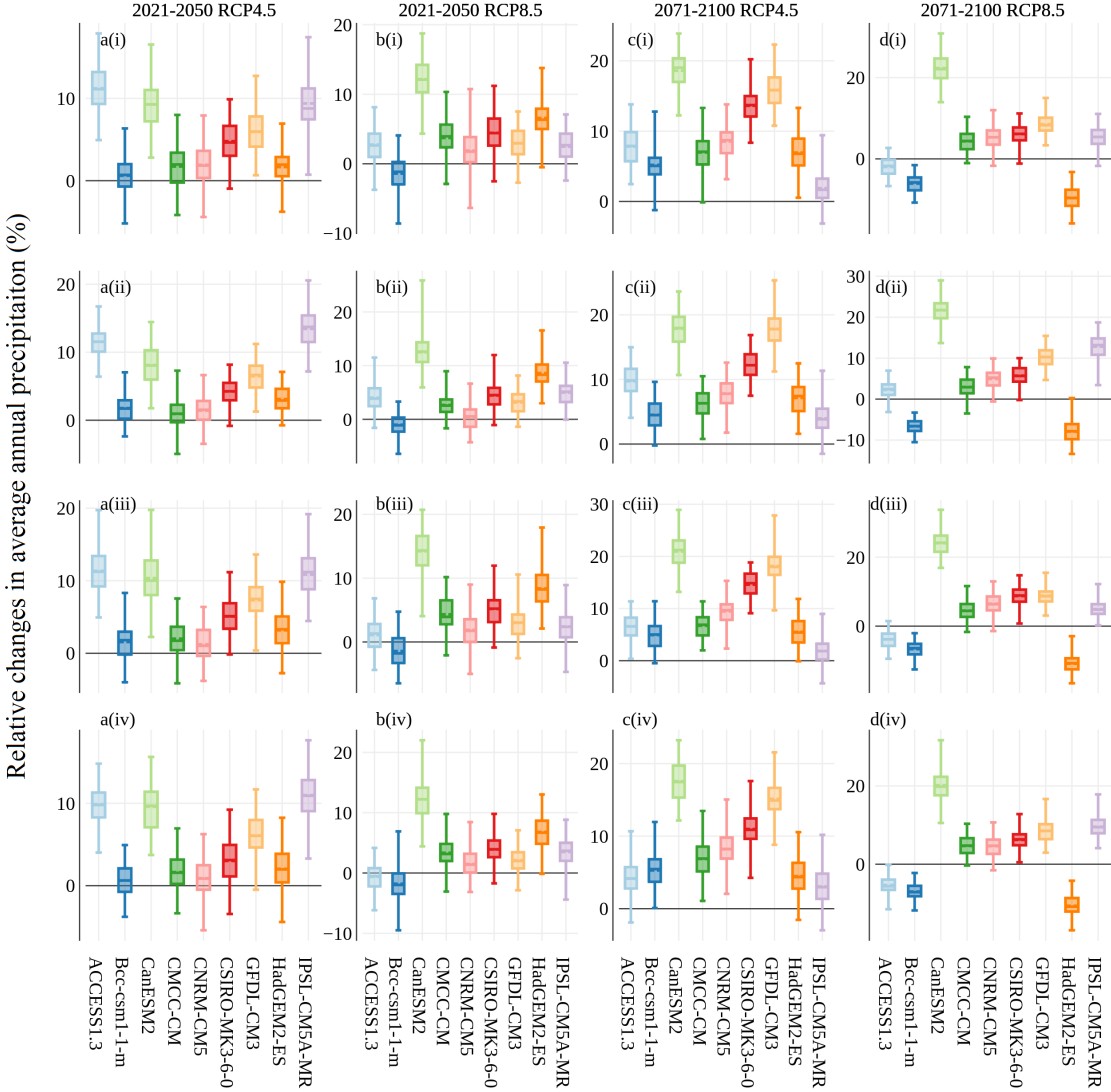

**Figure 4.** Relative changes (%) in long-term average annual precipitation from downscaled GCM data for: (**a**) 2021–2050 period under RCP4.5 scenario; (**b**) 2021–2050 period under RCP8.5 scenario; (**c**) 2071–2100 period under RCP4.5 scenario; and (**d**) 2071–2100 period under RCP8.5 scenario relative to observed values during 1985–2014. Rows (**i–iv**) denote results for stations S06, S23, S24, and S25, respectively. The bottom, middle, and top of the box denote the first, second, and third quartiles, respectively. The lower and upper ends of the whiskers represent the minimum and maximum values, respectively.

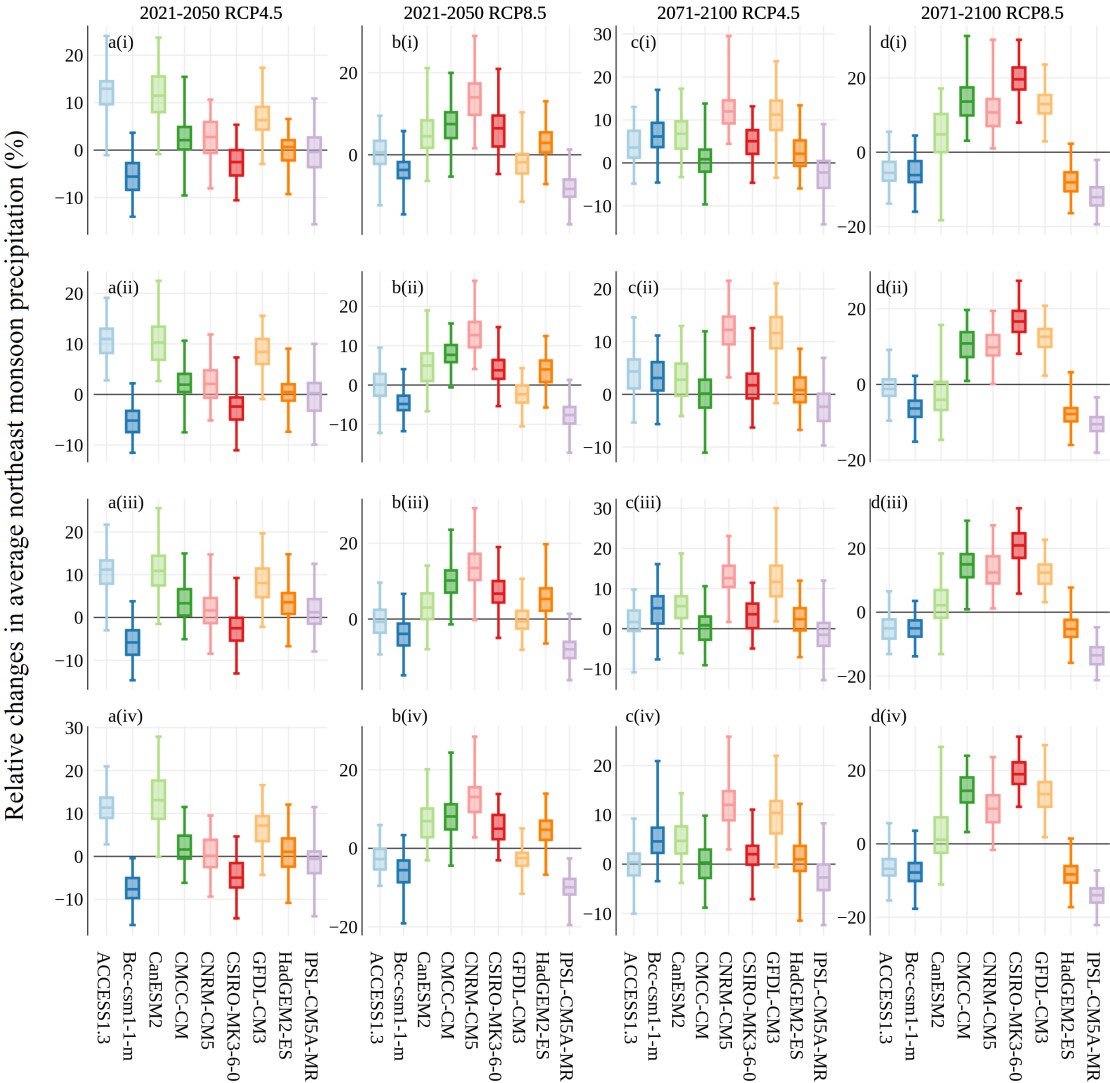

**Figure 5.** Relative changes (%) in long-term average northeast monsoon precipitation from downscaled GCM data for: (**a**) 2021–2050 period under RCP4.5 scenario; (**b**) 2021–2050 period under RCP8.5 scenario; (**c**) 2071–2100 period under RCP4.5 scenario; and (**d**) 2071–2100 period under RCP8.5 scenario relative to observed values during 1985–2014. Rows (**i**–**iv**) denote results for stations S06, S23, S24, and S25, respectively. The bottom, middle, and top of the box denote the first, second, and third quartiles, respectively. The lower and upper ends of the whiskers represent the minimum and maximum values, respectively.

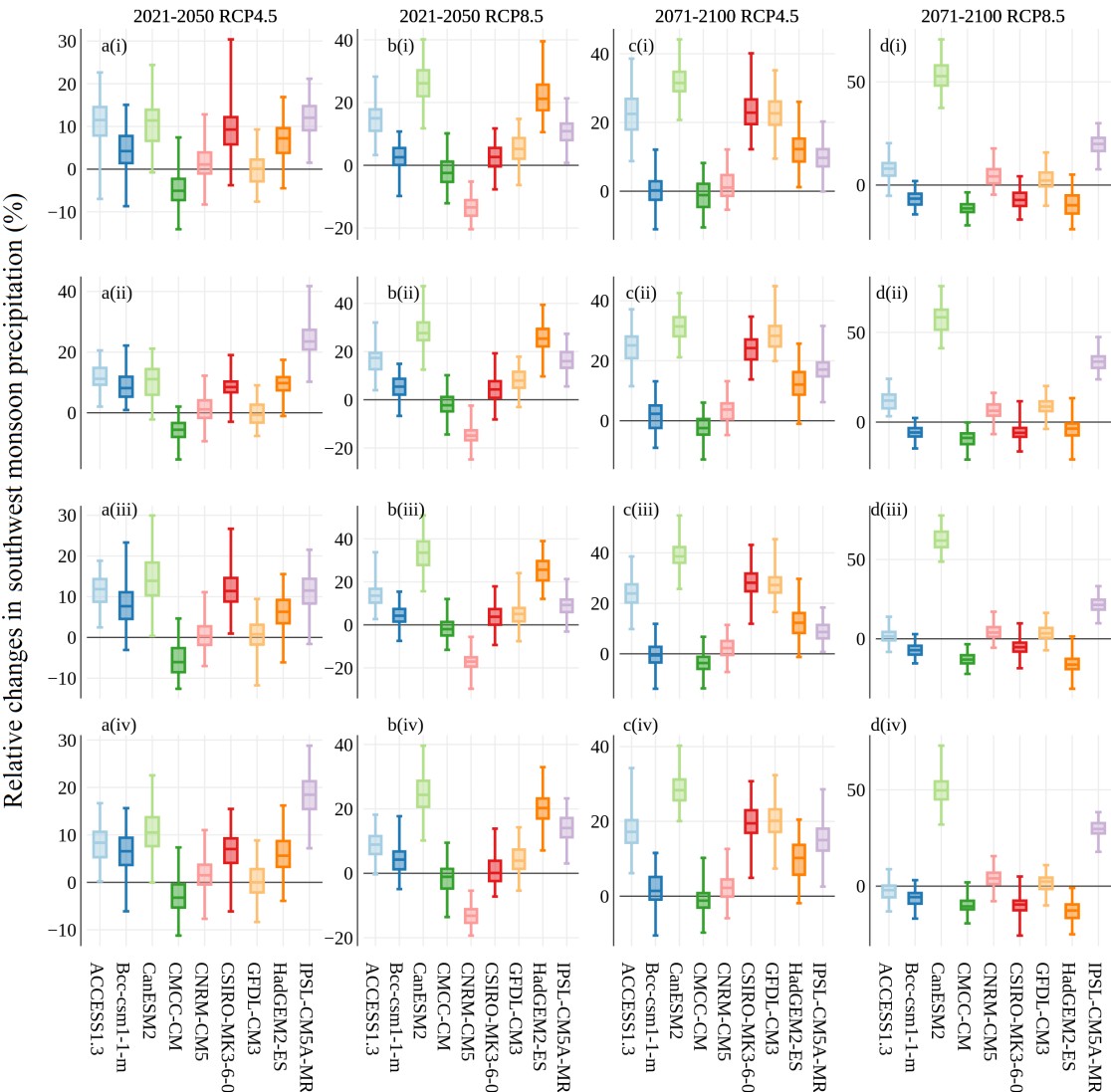

**Figure 6.** Relative changes (%) in long-term average southwest monsoon precipitation from downscaled GCM data for: (**a**) 2021–2050 period under RCP4.5 scenario; (**b**) 2021–2050 period under RCP8.5 scenario; (**c**) 2071–2100 period under RCP4.5 scenario; and (**d**) 2071–2100 period under RCP8.5 scenario relative to observed values during 1985–2014. Rows (**i–iv**) denote results for stations S06, S23, S24, and S25, respectively. The bottom, middle, and top of the box denote the first, second, and third quartiles, respectively. The lower and upper ends of the whiskers represent the minimum and maximum values, respectively.

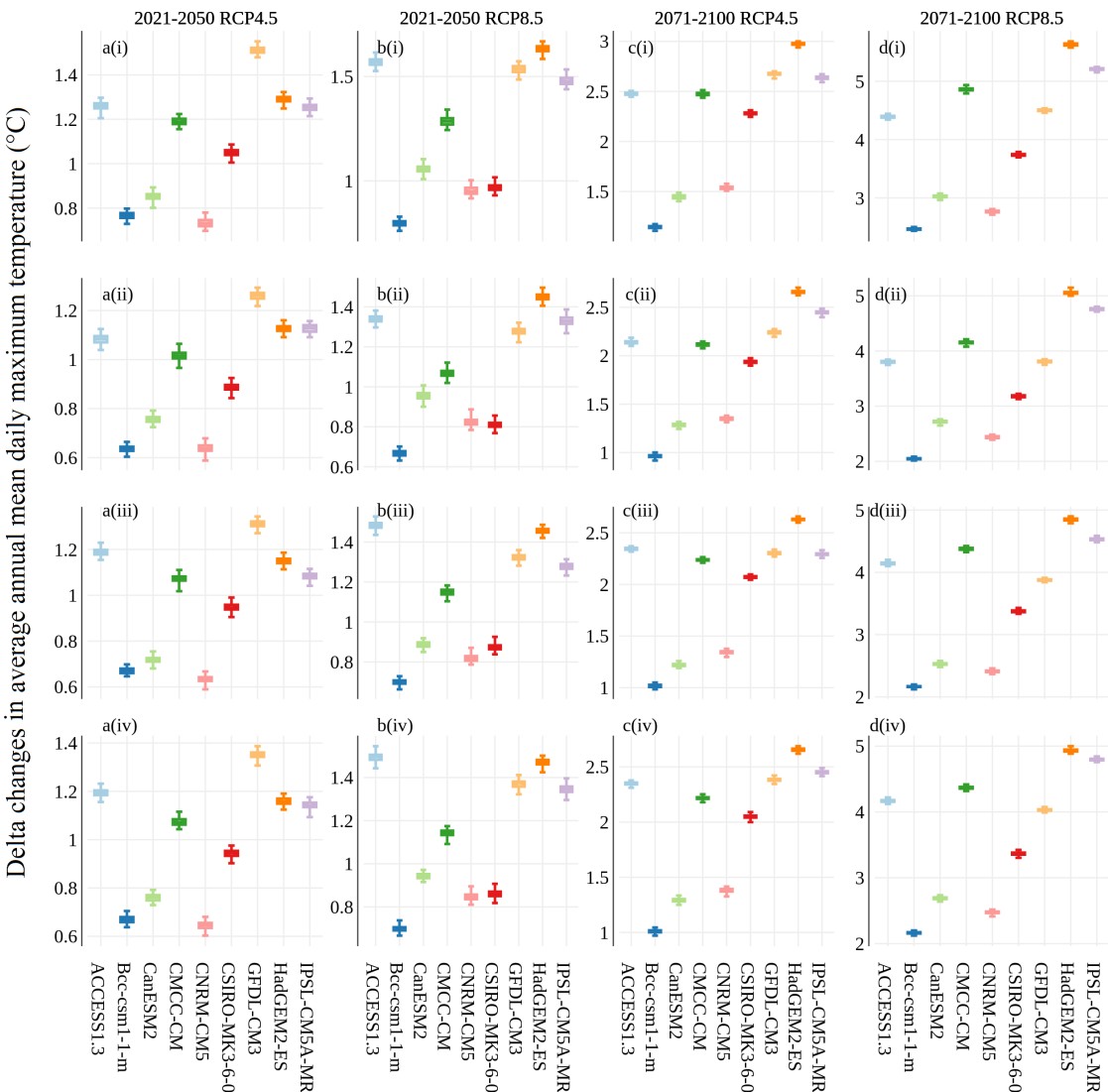

**Figure 7.** Delta changes in long-term average annual mean daily maximum temperature from downscaled GCM data for: (**a**) 2021–2050 period under RCP4.5 scenario; (**b**) 2021–2050 period under RCP8.5 scenario; (**c**) 2071–2100 period under RCP4.5 scenario; and (**d**) 2071–2100 period under RCP8.5 scenario relative to observed values during 1985–2014. Rows (**i–iv**) denote results for stations S06, S23, S24, and S25, respectively. The bottom, middle, and top of the box denote the first, second, and third quartiles, respectively. The lower and upper ends of the whiskers represent the minimum and maximum values, respectively.

It might be interesting to compare the change signal produced by the original GCM projections and by the downscaled data. Figure 9 presents the changes in mean monthly precipitation (% change) and mean monthly maximum and minimum temperatures (delta change) for the 2071–2100 period under RCP8.5 scenario relative to the observed values during the baseline period (1985–2014) for both the original GCM data and the downscaled data at station S24. As seen, for precipitation and maximum and minimum temperatures, the spatially downscaled data produce different change signals (in terms of both magnitude and direction of change) compared with the original GCM data. For example, we see less changes in average monthly precipitation in downscaled data compared with the original GCMs. For temperature, the downscaled data show consistent increase of average monthly maximum and minimum temperatures. In comparison, some GCMs show decrease of mean monthly maximum temperature for January, February, March, and April. The difference in change signals between original

GCM data and the downscaled ones indicate that climate change information might be different at different spatial scales. This has important implications for regional or local climate change studies.

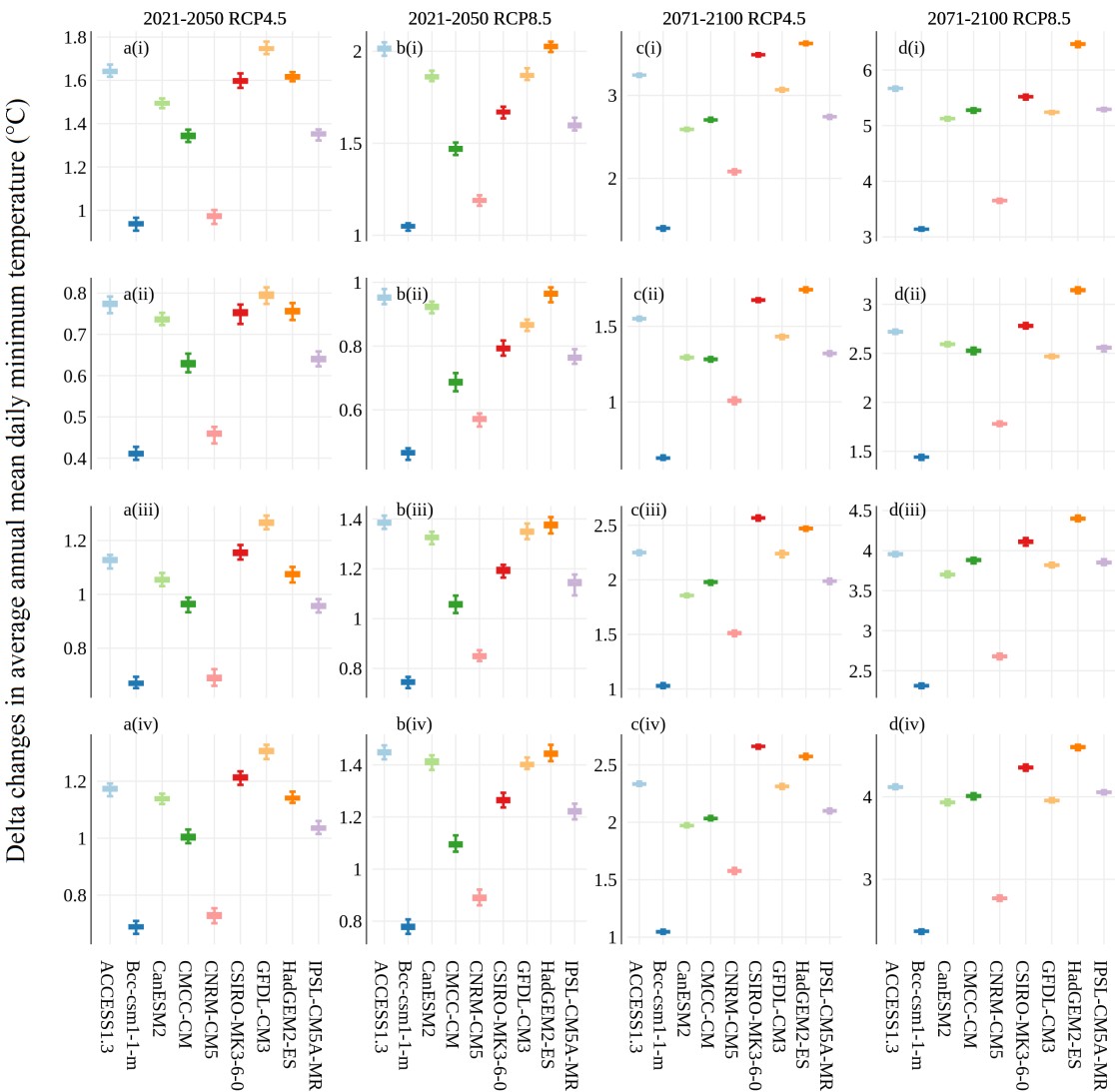

**Figure 8.** Delta changes in long-term average annual mean daily minimum temperature from downscaled GCM data for: (**a**) 2021–2050 period under RCP4.5 scenario; (**b**) 2021–2050 period under RCP8.5 scenario; (**c**) 2071–2100 period under RCP4.5 scenario; and (**d**) 2071–2100 period under RCP8.5 scenario relative to observed values during 1985–2014. Rows (**i–iv**) denote results for stations S06, S23, S24, and S25, respectively. The bottom, middle, and top of the box denote the first, second, and third quartiles, respectively. The lower and upper ends of the whiskers represent the minimum and maximum values, respectively.

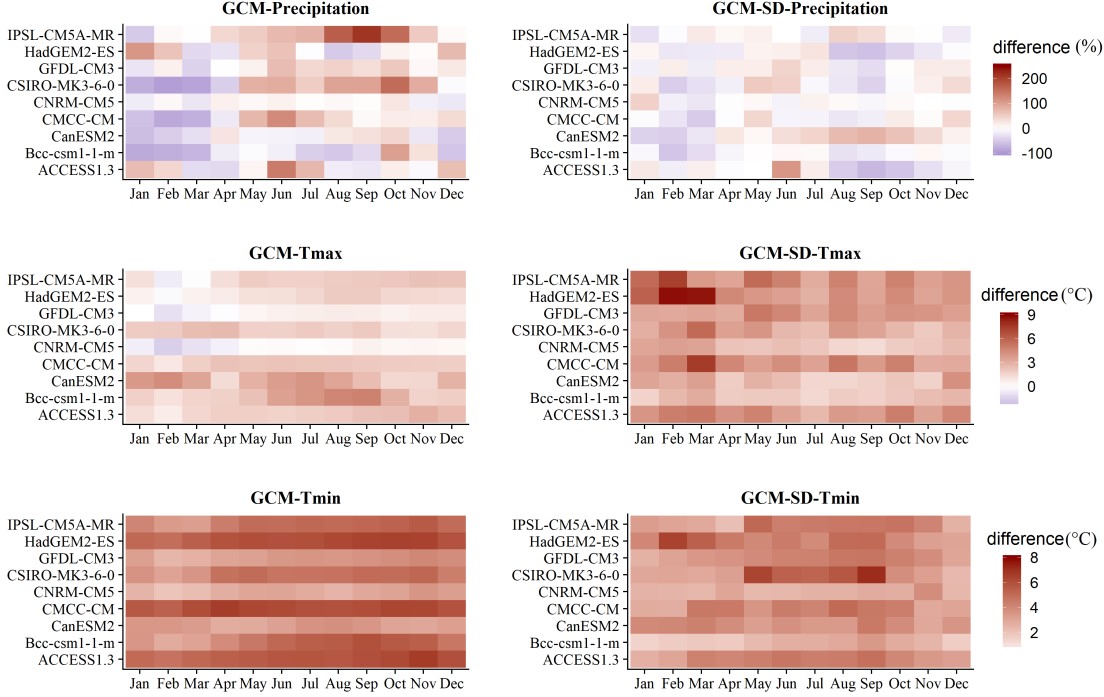

**Figure 9.** Relative Changes in mean monthly precipitation (first row,in %), mean monthly maximum temperature (second row; delta change), mean monthly minimum temperature (third row; delta change) for the 2071–2100 period under RCP8.5 scenario relative to the observed values during 1985–2014 at station S24. Left column denotes the original GCM values, right column represents the spatially and temporally downscaled GCM values.

In summary, the climate change signal over the Singapore region is more consistent for temperature than for precipitation. The mean annual average daily maximum and minimum temperatures are projected to increase over both the mid-century and end-century periods under both RCP4.5 and RCP8.5 emission scenarios. For precipitation, the projected changes vary across GCMs, RCPs, and projection periods at both annual and seasonal timescales. Similar results were found by Marzin et al. [28] based on a dynamical downscaling approach. As shown by Marzin et al. [28], there was a consensus on increase of annual average daily maximum and minimum temperatures, with increase becoming greater from RCP4.5 scenario to RCP8.5 scenario, and from mid-century period to end-century period; however, for precipitation, as shown by Marzin et al. [28], disagreement was observed among the GCMs, and both positive and negative changes in annual and seasonal precipitation could happen, which are independent of emission scenarios.

### 4.2.2. Changes in Average Wet and Dry Spell Lengths

The day-to-day variability of precipitation can be characterized by wet and dry spells. The change in wet and dry spell characteristics has great impact on hydrology, agriculture, ecology and water resources management. A wet (dry) spell is defined as the number of consecutive rainy (non-rainy) days (a rainy day is defined as the day with daily precipitation ≥ 1 mm). A number of dry and wet spell indices (e.g., see Li et al. [26] and Li et al. [49]) could be utilized to analyze the characteristics of dry and wet spells; however, here, due to the space limit, we only focus on the changes in average dry and wet spell length. Figures 10 and 11 present, respectively, the relative changes in average dry and wet spell lengths with respect to the observed values during 1985–2014. It is necessary to emphasize here that the EC-component of the multi-site multivariate downscaling approach plays a significant role in building the wet and dry spell characteristics. As seen, most of the GCMs tend to project positive median change values for average dry spell length over both the 2021–2050 and 2071–2100 periods under both emission scenarios. This indicates that average dry spell length is more

likely to increase over the 2021–2050 and 2071–2100 periods under both RCP4.5 and RCP8.5 scenarios. Compared with average dry spell length, less consensus is observed about the projected changes in average wet spell length. Overall, average dry spell length is projected to change by −7.3–11.4% and −4.7–21.4% over the 2021–2500 period under RCP4.5 and RCP8.5 scenarios, respectively. For the period of 2071–2100, the spreads of the projected changes in average dry spell length are −7.5–11.7% and −3.8–28.1% under RCP4.5 and RCP8.5 scenarios, respectively. For average wet spell length, the ranges of the projected changes under RCP4.5 and RCP8.5 scenarios, respectively, are −5.6–17.1% and −8.9–11.0% over the 2021–2050 period, and −5.4–17.2% and −13.4–11.9% over the 2071–2100 period. These projected changes have implications for adaptation planning and risk reduction in water resources management in Singapore.

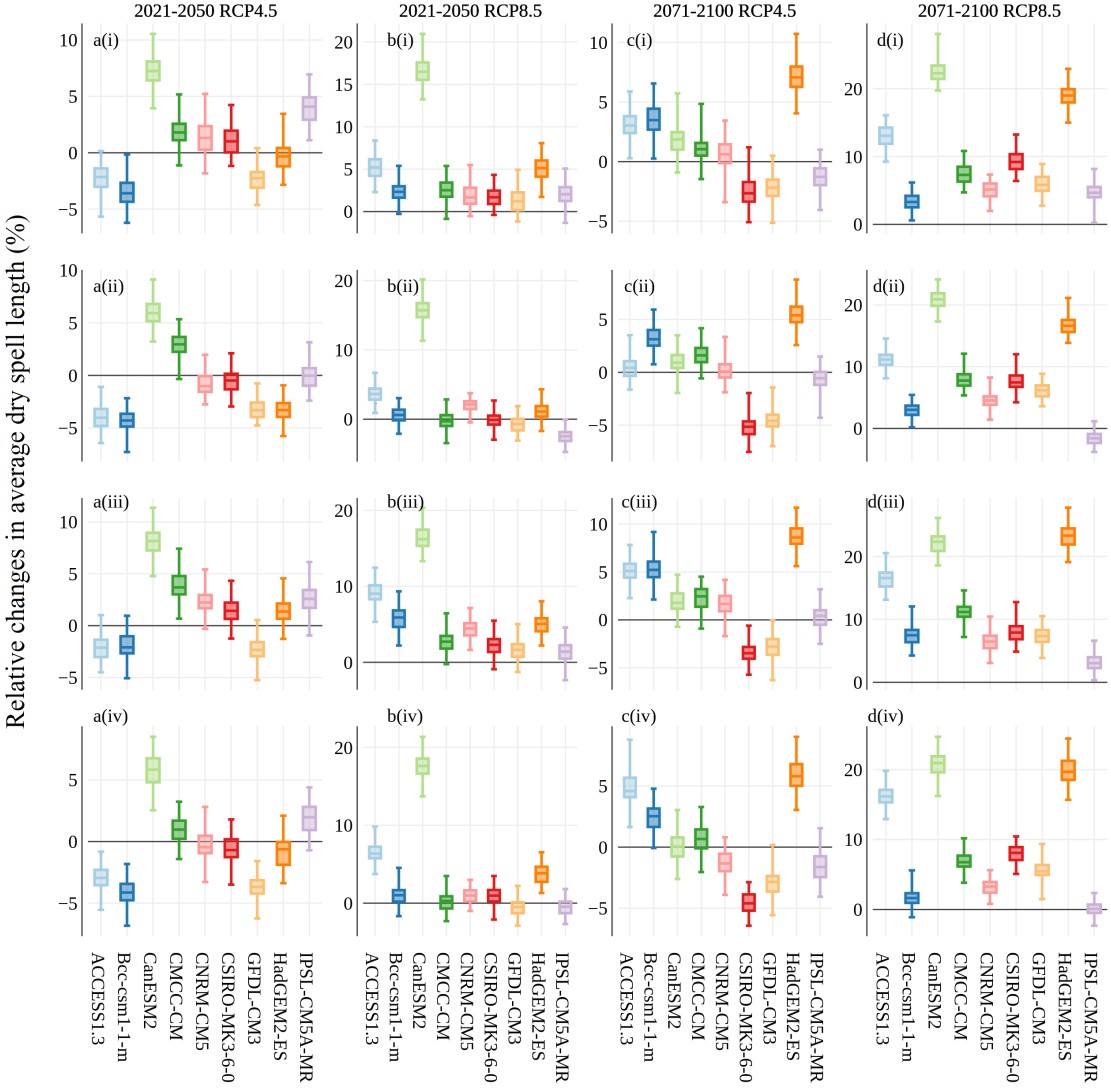

**Figure 10.** Relative changes (%) in average dry spell lengths from downscaled GCM data for: (**a**) 2021–2050 period under RCP4.5 scenario; (**b**) 2021–2050 period under RCP8.5 scenario; (**c**) 2071–2100 period under RCP4.5 scenario; and (**d**) 2071–2100 period under RCP8.5 scenario relative to observed values during 1985–2014. Rows (**i–iv**) denote results for stations S06, S23, S24, and S25, respectively. The bottom, middle, and top of the box denote the first, second, and third quartiles, respectively. The lower and upper ends of the whiskers represent the minimum and maximum values, respectively.

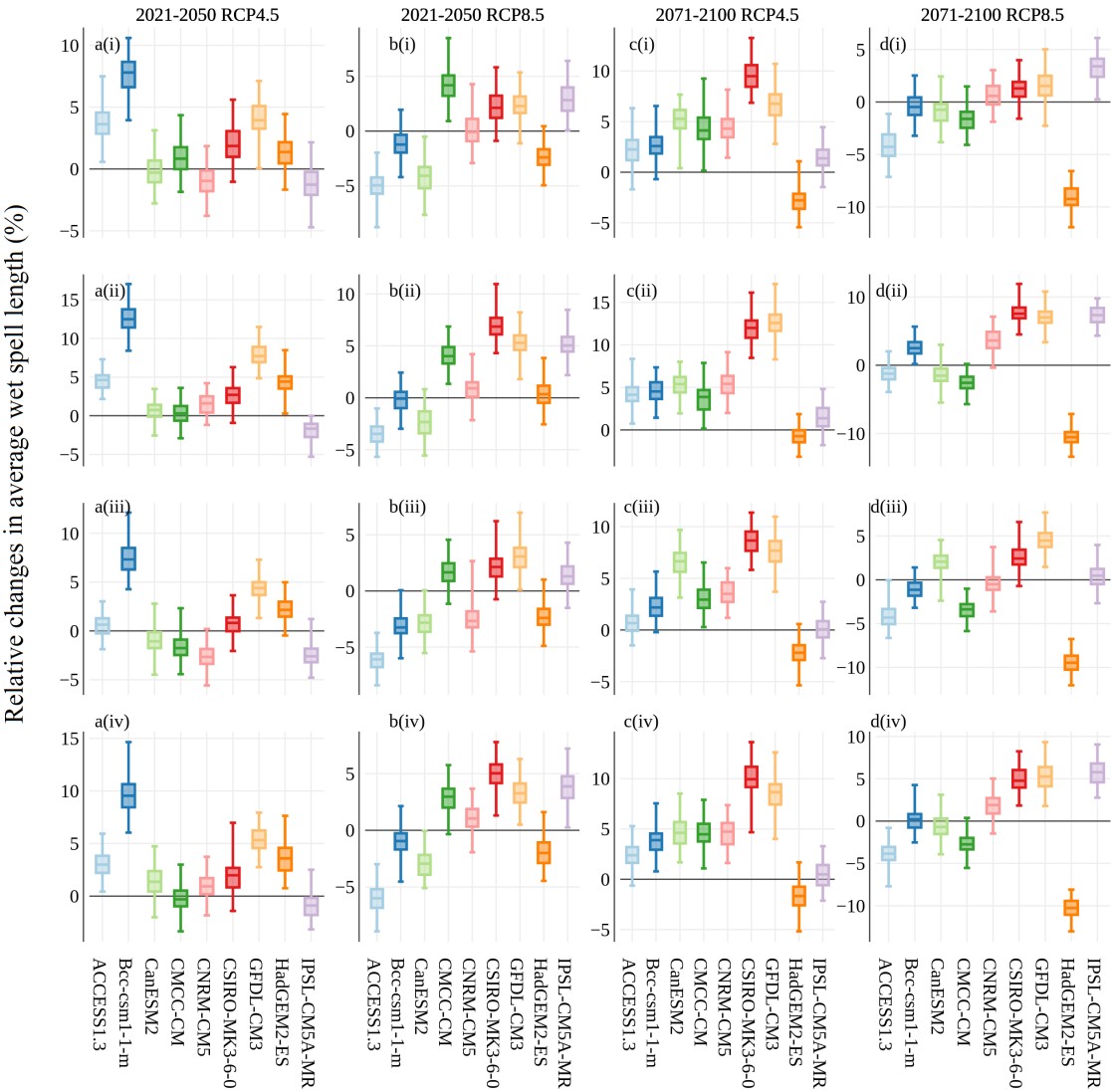

**Figure 11.** Relative changes (%) in average wet spell lengths from downscaled GCM data for: (**a**) 2021–2050 period under RCP4.5 scenario; (**b**) 2021–2050 period under RCP8.5 scenario; (**c**) 2071–2100 period under RCP4.5 scenario; and (**d**) 2071–2100 period under RCP8.5 scenario relative to observed values during 1985–2014. Rows (**i**–**iv**) denote results for stations S06, S23, S24, and S25, respectively. The bottom, middle, and top of the box denote the first, second, and third quartiles, respectively. The lower and upper ends of the whiskers represent the minimum and maximum values, respectively.

#### 4.2.3. Uncertainty of the Projection Results

The projection results presented in this paper may be subject to a cascade of uncertainties resulting from several factors. The first one is related to the choice of GCMs. In the present study, only nine GCMs were used for downscaling, and, for each GCM, only the first realization (r1i1p1) was used. As such, the full range of potential climate change as well as the impact of the initialization of GCMs (e.g., the impact of the different atmospheric conditions used at the start of each experiment) on the projection results might not be well represented. Second, we only used two representative concentration pathways, i.e., the RCP4.5 and RCP8.5 scenarios, as the prescribed future emission scenarios, which may not cover the entire set of possible greenhouse gas emission scenarios. Third, all the large-scale GCM outputs were downscaled by using only one statistical downscaling method, i.e., the multi-site, multivariate downscaling approach of Li and Babovic [19]; therefore, uncertainty due to the choice of downscaling techniques can also be expected.

In this study, the uncertainties were linked to GCMs, RCP scenarios, and the stochastic variability of the multi-site multivariate downscaling method. As the last one could be clearly represented by the range of the boxes from the 100 realizations, we place our focus on the uncertainties resulted from GCMs and RCPs. Figure 12 shows the relative changes (%) in average annual precipitation versus the delta changes in average annual mean daily maximum temperature for different GCMs and RCP scenarios for the 2071–2100 period at station S06. Since, for each GCM and RCP, there are 100 realizations of projections, to put aside the uncertainty from the stochastic variability of the downscaling method, the median values of the 100 realizations are shown. As seen, the uncertainties related to the GCMs and RCPs are evident. For instance, six GCMs (in the vertical blue rectangle in Figure 12) project similar increases of temperature but different changes in precipitation under RCP4.5 scenario. Similarly, four GCMs (in the horizontal red rectangle in Figure 12) project similar changes in precipitation but different increases of temperature under RCP8.5 scenario. The projected directions of change in precipitation differ even for different RCPs (for instance, the median change values for ACCESS1.3, Bcc-csm1–1-m, and HadGEM2-ES under RCP 4.5 scenario differ in direction from RCP8.5 scenario; see Figure 4c(i),d(i) for details). Overall, the uncertainty ranges of RCP8.5 scenario is larger than those of RCP4.5 scenario, for both precipitation and temperature.

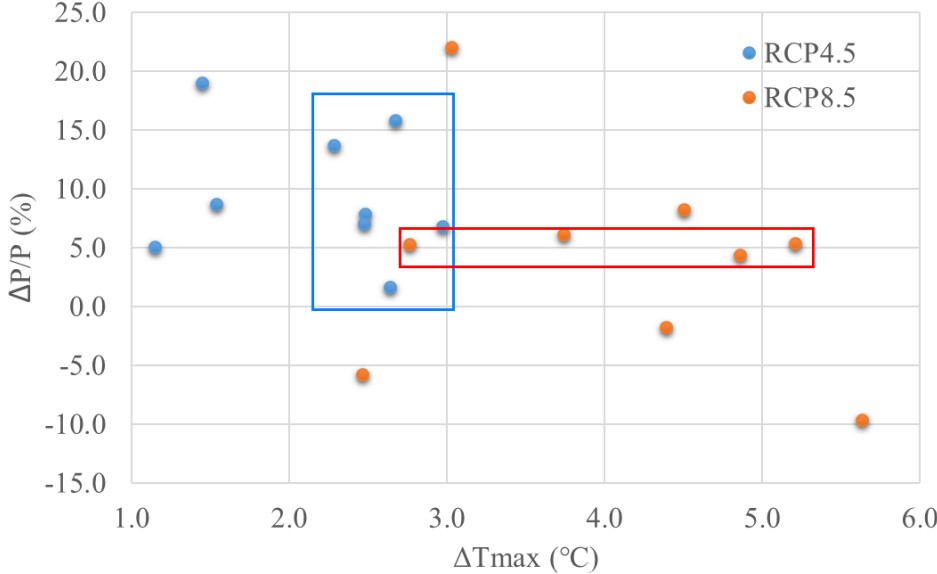

**Figure 12.** Projected relative changes (%) in average annual precipitation and delta changes (°C) in average annual mean daily maximum temperature for different GCMs and RCPs for the 2071–2100 period at station S06. The median value of the 100 realizations is shown for a given GCM and RCP.

With respect to the multi-site, multivariate downscaling approach itself, the assumptions involved in the different stage of downscaling also have impacts on the projections results. For example, for the quantile mapping approach used in the spatial downscaling, the regression relationship fitted between the GCM grid-scale monthly data and the local station-specific ones are assumed to be stationary. Based on such an assumption, Mullan et al. [36] demonstrated the capability of the quantile mapping approach in reproducing the observed distributional statistics in a non-stationary climate condition. However, it could also be expected that the regression weights fitted based on historical data would change through time, leading to a deterioration of performance for quantile mapping over a future period. Besides, the non-stationarity of the GCM biases could also affect the performance of quantile mapping. Another important assumption made in the downscaling approach is the stationarity of EC for post-processing the spatially-temporally downscaled GCM outputs. Whether this assumption is valid under a non-stationary climate condition determines the performance of the proposed multi-site multi-variate downscaling approach in representing the various forms of dependencies. One may

argue that this assumption might be more valid during the mid-century period than during the end-century period, given that climate condition is more likely to change in the distant future than in the near future. However, this was not considered in the present study, since we assumed the stationarity of EC throughout the future period.

## 5. Conclusions

A reliable estimate of the projected changes in precipitation and temperature has great significance for subsequent impact modeling and adaption planning in the context of climate change. To generate spatially-distributed and physically coherent climate projections at regional scale, multi-site and multivariate downscaling approaches are often desired to reflect the inter-site and inter-variable dependence structures in the downscaled meteorological field. In this study, mid-term and long-term projections of precipitation and maximum and minimum temperatures at four weather stations of the Singapore region were generated using a newly developed multi-site and multivariate downscaling approach. This downscaling approach was evaluated fully by Li and Babovic [19].

The projections show that average maximum and minimum temperatures in Singapore would increase in the future, while annual precipitation may increase or decrease, depending on the GCMs and emission scenarios. The major uncertainty of precipitation projection comes from GCMs, which implies that an ensemble of GCMs is needed for impact assessment in this region. The projected changes in precipitation and temperature in Singapore may affect various aspects of the natural environment, public heath, energy demand, and urban infrastructure. To cope with the projected climate changes, appropriate adaption strategies should be formulated and they should be flexible to incorporate future knowledge of climate change to safeguard the nation against the potential risks.

The multi-site multivariate downscaling method used in this study can: (1) provide a distributed, physically coherent downscaled meteorological fields for subsequent impact modeling in different fields; and (2) generate an ensemble of realizations of climate change scenarios for better characterization of the uncertainty in the era of global warming. Future research should focus on the coupling of this method with hydrological models, to analyze the climate change impact on hydrological variabilities. Moreover, comparison studies with other downscaling approaches across diverse climate and topographies would also be interesting.

**Author Contributions:** Conceptualization, X.L.; methodology, X.L. and K.Z.; formal analysis, X.L., K.Z., and V.B.; investigation, X.L. and K.Z.; writing—original draft preparation, X.L.; writing—review and editing, X.L., K.Z., and V.B.; visualization, X.L.; funding acquisition, K.Z. and X.L.

**Funding:** This research was funded by National Natural Science Foundation of China grant number 51909061, Natural Science Foundation of Jiangsu Province grant number BK20180022, Fundamental Research Funds for the Central Universities grant number 2019B01614, and Six Talent Peaks Project in Jiangsu Province grant number NY-004.

**Conflicts of Interest:** The authors declare no conflict of interest.

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
