# Peer review of "Projections of Future Climate Change in Singapore Based on a Multi-Site Multivariate Downscaling Approach"

_water, doi:10.3390/w11112300_

Round 1
Reviewer 1 Report
Thank you for the opportunity to review the manuscript. The paper is interesting and will undoubtedly attract the interest of a broad international audience. There is likely merit in publishing the results and presenting its conclusions to the broad audience, but I believe the manuscript needs some restructuring. First, it seems that the text is a bit too long, which makes it difficult to follow. Moreover, the nature of the presented material indicates that the classic division into Result and Discussion sections should be maintained. Conclusions resemble a summary of the text. Please reformulate the conclusion of your study more compellingly: provide a clear summary of how your study contributes to new knowledge or understanding about the problem, synthesize the key points, and recommend areas for future research.
Specific comments:
Lines 169 -196 please move all this paragraph to the methods section
Lines 333-335 this sentence is unclear
Page 22 presentation of the temperature projection with precision to two decimal places is probably incorrect and not very reasonable.
Some projection results of dry and wet periods are not convincing. How to explain the result given on lines 513-514: "dry spell length is projected to change by -7.3-11.4%"?
Section 4.2.3 I would like to see the results of the uncertainty analysis presented in a numerical and not just a descriptive way.
Please provide a few sentences on the importance of your results for the environment and business practice.
Fig. 1 in the legend of the figure, altitude above sea level is given in the range 185 to -43 m. Is this depression only a result of extrapolation? Please make the given values ​​real.
Reviewer 2 Report
The present manuscript is an interesting study focused on the downscaling techniques that have to be used to increase the spatial and temporal resolutions of the projections provided by Global Climate Models. This problem is very important as climatic trends can have distinct impacts on natural environments and human activities at local scales. The manuscript is well written and clearly presented, but some figures might be improved. I suggest only the following minor changes, before its publication on Water:
Page 8 lines 151-153: this sentence is unnecessary and it might be deleted.
Page 8 line 158: please, add the link to Figure 1 here, at the end of the first sentence of the Section 2.
Figure 1: The map shown in Figure 1 might be improved, distinguishing with distinct colors/lines the urbanized areas of Singapore from the natural areas. In fact, most of the green area shown in the map is not natural, but it is occupied by the city. The presence of urbanized areas affects the values of meteorological parameters, like the air temperature.
Page 11 lines 218-219: The following sentence is also unnecessary and it could be deleted: “The basic procedure of implementing this method is illustrated as follows”.
Page 12 lines 240-241: Please, check the meaning of this sentence. It should be: “the normal distribution of Li et al. (2017) was preferred, being it simpler than the first-order autoregressive model of Richardson (1981).”
Page 12 line 242: delete “of”
Page 12 line 247-249: For downscaling of the precipitation, parameters need to be determined for both the precipitation occurrence model (parameters being two transition probabilities) and precipitation amount model.
Pages 21-22: The numeration of some figures do not follow the appearance in the text: figures 8 and 9 are discussed before figure 7. Figure numbers 8, 9, 7 should be renumerated as 7, 8 and 9, respectively.
Figures 4-11, Pages 38-40. Please, show Y-axes in the panels of Box Whisker Plots (Figures 4, 5, 6, 8, 9, 10, 11). In each row, the first panel on the left can show the axis, tick marks and tick numbers. If Y-scale does not change, the other panels of the row can show only the axis and tick marks.
Round 2
Reviewer 1 Report
I recommend the revised manuscript for publication